# Full-spectrum nonmetallic plasmonic carriers for efficient isopropanol dehydration

Changhai Lu[1,4], Daotong You[2,4], Juan Li[1], Long Wen[1], Baojun Li [1],
Tuan Guo [2,3] ✉ & Zaizhu Lou [1] ✉

Plasmonic hot carriers have the advantage of focusing, amplifying, and manipulating optical signals via electron oscillations which offers a feasible pathway to influence catalytic reactions. However, the contribution of non-metallic hot carriers and thermal effects on the overall reactions are still unclear, and developing methods to enhance the efficiency of the catalysis is critical. Herein, we proposed a new strategy for flexibly modulating the hot electrons using a nonmetallic plasmonic heterostructure (named $W_{18}O_{49}$-nanowires/reduced-graphene-oxides) for isopropanol dehydration where the reaction rate was 180-fold greater than the corresponding thermocatalytic pathway. The key detail to this strategy lies in the synergetic utilization of ultraviolet light and visible-near-infrared light to enhance the hot electron generation and promote electron transfer for C-O bond cleavage during iso-propanol dehydration reaction. This, in turn, results in a reduced reaction activation barrier down to 0.37 eV (compared to 1.0 eV of thermocatalysis) and a significantly improved conversion efficiency of 100% propylene from iso-propanol. This work provides an additional strategy to modulate hot carrier of plasmonic semiconductors and helps guide the design of better catalytic materials and chemistries.

As the second-largest petrochemical after ethylene, propylene is widely used in many fields, especially in the form of polypropylene used in the medical field[1,2]. In industry, propylene is manufactured via steam cracking of light naphtha and fluid catalytic cracking of natural gas or coal with high energy consumption[3,4]. However, the nonrenewable nature of fossil fuels has forced people to explore the production of propylene from a sustainable source (e.g., biomass)[5–7]. For example, propylene can be produced from bioethanol via dimerization and metathesis processes at high temperature (673–773 K), with $Al_2O_3$, zeolites, nickel (II) 2-iminopyridine, etc., a as catalysts[8,9]. With the development of carbohydrate fermentation for bioisopropanol production[10], a highly selective isopropanol dehydration reaction is more attractive for propylene production. Gallium borates with

Brønsted acid sites were reported to show 98.3% conversion of iso-propanol to 100% propylene at 573 K[11]. Chen et al. studied the active sites on the $WO_x$/Pt (111) surface for the dehydration and dehydrogenation of isopropanol, demonstrating dominant dehydration reaction on $W_3O_9$ clusters with an activation barrier of 1.23 eV[12]. However, all these reports showed that a high temperature was required to active the catalysts and that the reactions were associated with high activation barriers. It is imperative to develop miniature operated solutions under mild conditions and to demonstrate highly efficient and selective isopropanol dehydration for propylene production.

Metal nanoparticles with surface plasmon resonance (SPR) can generate highly active hot electrons for chemical reactions[13–16]. Halas et al. quantified the effects of the metallic plasmonic carrier and

[1]Institute of Nanophotonics, Jinan University, Guangzhou 511443, China. [2]Institute of Photonics Technology, Jinan University, Guangzhou 511443, China. [3]Southern Marine Science and Engineering Guangdong Laboratory (Zhuhai), Zhuhai 519000, China. [4]These authors contributed equally: Changhai Lu, Daotong You. ✉e-mail: tuanguo@jnu.edu.cn; zzlou@jnu.edu.cn

photothermal on the promotion of catalytic ammonia decomposition via Cu-Ru, verifying the dominant role of hot carriers in boosting the reaction[17]. Compared to noble metals, low-cost semiconductors with heavy doping also exhibit SPR in the visible and near-infrared (NIR) regions. Moreover, their abundant surface vacancies provide many active sites for chemical reactions[18–21]. For example, plasmonic $Bi_2WO_6$ with oxygen vacancies close to W atoms exhibits SPR-dependent methane generation during the $CO_2$ reduction reaction[18]. $WO_{3-x}$ nanowires (NWs) with strong SPR in the visible-NIR spectral region have been used to promote the Suzuki coupling reaction, hydrogen generation and $CO_2$ reduction[22–25]. Recently, dehydration and dehydrogenation of ethanol were realized by using plasmonic $WO_{3-x}$ as a catalyst driven by solar energy[26,27], and Ma et al. constructed $WO_{3-x}$/carbon hybrids to enhance bioethanol dehydration for ethylene[28]. However, the contribution of nonmetallic hot carriers and thermal effects on catalysis are still unclear, and how to enhance the catalytic efficiency of nonmetallic plasmonic carriers remains a challenging task.

Herein, we propose a new strategy for flexibly modulating hot electrons using a nonmetallic plasmonic heterostructure synthesized by controlling $W_{18}O_{49}$-NW growth on reduced graphene oxides (rGO), in which the rGO can stabilize the surface oxygen vacancies of the plasmonic $W_{18}O_{49}$-NWs to achieve strong SPR and provide more active sites for photocatalytic isopropanol dehydration. Under full-spectrum irradiation, ultraviolet light excited electron accumulation and visible–NIR-excited SPR on heterostructure $W_{18}O_{49}$-NWs/rGO have a synergy on enhancing hot electron generation, and which has been demonstrated to play a dominant role in boosting isopropanol dehydration by promoting electron transfer to break C–O bond in the transition stare during dehydration, thus reducing the reaction activation barrier to 0.37 eV (compared to 1.0 eV for thermocatalysis). Therefore, the optimal $W_{18}O_{49}$-NWs/rGO-1% can catalyze isopropanol dehydration to achieve nearly 100% propylene with a rate of 437 mmol $g^{-1} h^{-1}$, which is over 180-fold higher than that of thermocatalysis. This work provides a crucial strategy to modulate the nonmetallic hot carriers of plasmonic semiconductor and helps guide the design of better catalytic chemistries.

## Results

### Structural and optical properties of nonmetallic plasmonic heterostructures

Tungsten oxides $W_{18}O_{49}$ with abundant oxygen vacancies and SPR were chosen to construct the plasmonic heterostructures for catalysis. rGO layers[30] and plasmonic $W_{18}O_{49}$-NWs[31] with a diameter of 10–15 nm and a length of 1.5–2 μm were synthesized as references (Fig. 1a, b). A GO solution was mixed with a $WCl_6$ precursor in ethanol. After solvothermal treatment, rGO with tungsten oxide grew on the surface as plasmonic heterostructures. Figure 1c, d shows the morphologies of these heterostructures as fine NWs (5 nm in diameter) dispersed uniformly on rGO layers. The high-resolution transmission electron microscopy (HRTEM) image (Fig. 1e) shows a clear lattice spacing of 0.38 nm for the NWs (blue area), which was assigned to the (010) facets of $W_{18}O_{49}$[29]. Moreover, the disordered atomic arrangements (orange area) indicate abundant oxygen vacancies on the surface. The morphologies of $W_{18}O_{49}$-NWs/rGO with different compositions are shown in Fig. S1. As the content of rGO increased, the $W_{18}O_{49}$-NWs became uniformly dispersed on the rGO layers, and their surface became covered by rGO. Further analysis using high-angle annular dark-field STEM (HAADF-STEM) (Fig. 1f) shows clear NWs loading on the rGO layers. EDS elemental mapping images

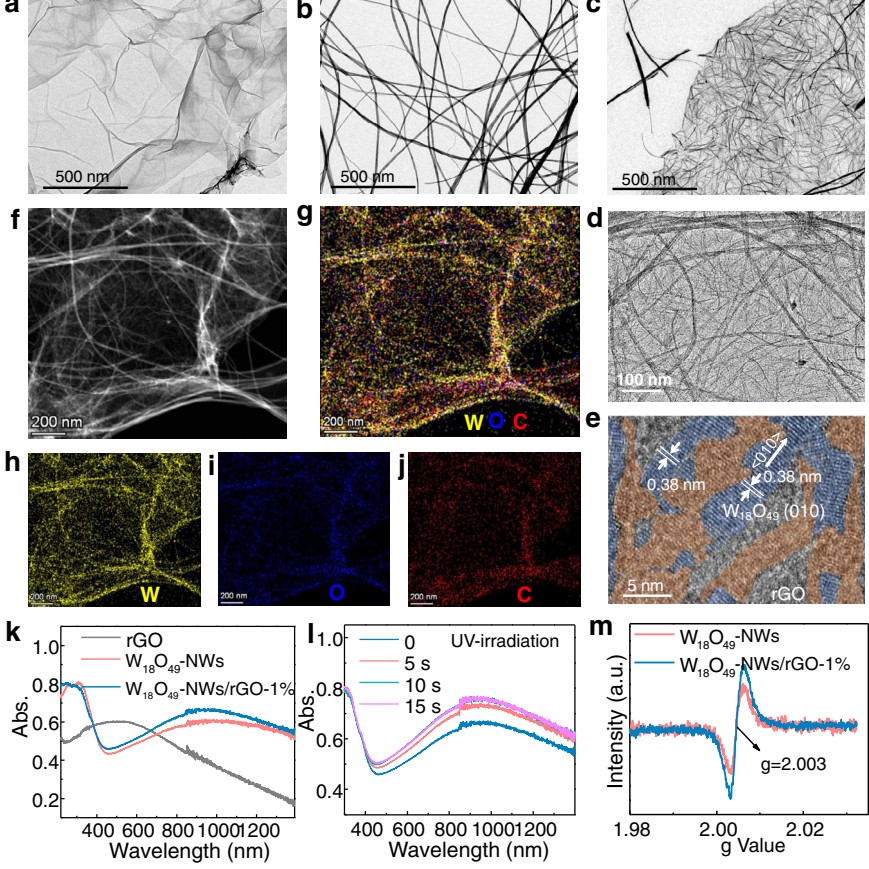

**Fig. 1 | Characterization of catalysts with various analysis techniques. a, b** TEM images of rGO and $W_{18}O_{49}$-NWs. **c, d** TEM, **e** HRTEM, **f** HAAD-STEM and **g–j** EDS elements (W, O, C) mapping images of $W_{18}O_{49}$-NWs/rGO-1% heterostructures. **k** UV–visible–NIR DRS of rGO, $W_{18}O_{49}$-NWs and $W_{18}O_{49}$-NWs/rGO-1%. **l** UV irradiation-induced DRS varies of $W_{18}O_{49}$-NWs/rGO-1%. **m** EPR spectra of $W_{18}O_{49}$-NWs and $W_{18}O_{49}$-NWs/rGO-1%.

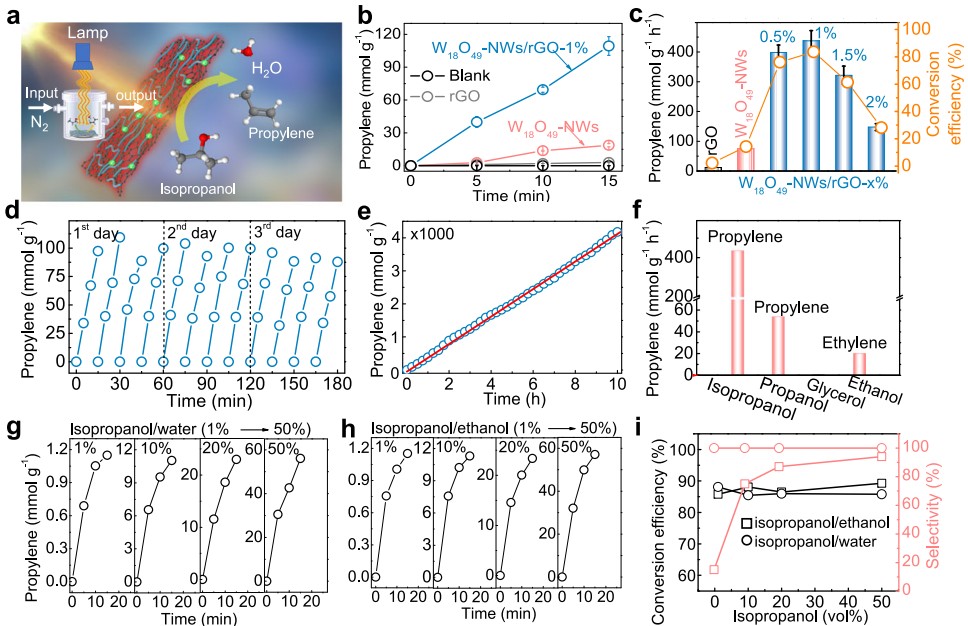

**Fig. 2 | Photocatalytic isopropanol dehydration to propylene. a** Diagram depicting photocatalytic isopropanol dehydration in a reaction chamber. **b** Propylene generation from isopropanol dehydration over rGO, $W_{18}O_{49}$-NWs, $W_{18}O_{49}$-NWs/rGO-1% and without catalysts under full-spectrum irradiation. **c** Propylene generation rates over various $W_{18}O_{49}$-NWs/rGO heterostructures. Bars represent mean values ± SD. **d** Twelve cycles of isopropanol dehydration reactions over 3 days. **e** Long term continuous isopropanol dehydration reaction for propylene (10 h). **f** Photocatalytic performance of $W_{18}O_{49}$-NWs/rGO-1% in several alcohol dehydration reactions. **g, h** Isopropanol dehydration from isopropanol/water and isopropanol/ethanol mixtures with various ratios, and **i** their conversion efficiency and selectivity. The irradiation source was full-spectrum (200–1100 nm) light with an intensity of 200 mW/cm² supplied by a Xenon lamp.

(Fig. 1g, j) show that the W and O atoms were mainly located on NWs, while, the C atoms were mainly distributed on the layers. X-ray diffraction patterns (Fig. S2) show the crystal structures of the $W_{18}O_{49}$-NWs. Their Raman spectra (Fig. S3) show two peaks located at 1580 and 1350 cm⁻¹ corresponding to the G and D bands of rGO[32]. The G band was assigned to the vibration of $sp^2$-bonds, while, the D band was correlated with defects derived from grain boundaries and vacancies[33]. The $I_D/I_G$ ratio was calculated to be 0.86, 1.00, and 1.15 for GO, rGO, and $W_{18}O_{49}$-NWs/rGO-1%, respectively, implying more oxygen vacancies in the heterostructure[34].

The optical properties of the $W_{18}O_{49}$-NWs/rGO heterostructure were investigated by UV–visible–NIR diffuse reflectance spectroscopy (DRS, Fig. 1k). The $W_{18}O_{49}$-NWs exhibited strong SPR in the visible-NIR spectral region, caused by the electron oscillation[35]. The DRS of $W_{18}O_{49}$-NWs/rGO-1% was enhanced in the visible region due to the light absorption of rGO, furthermore, their SPR band was enhanced and slightly blueshifted because of the greater number of oxygen vacancies. As the rGO content increased from 0.5 to 2 wt%, the heterostructures showed a stronger SPR band (Fig. S4). More rGO covered the $W_{18}O_{49}$-NWs, stabilizing the surface oxygen vacancies to generate strong SPR but restricting surface active site exposure for catalysis. Under UV light irradiation, their SPR band became stronger and slightly blueshifted, and the strongest intensity was observed at 10 s (Fig. 1l). These results indicate that UV-excitation can cause electron accumulation for SPR modulation, and which can facilitate hot electron generation. The electron paramagnetic resonance (EPR) spectra (Fig. 1m) showed that more oxygen vacancies were present on the plasmonic heterostructure $W_{18}O_{49}$-NWs/rGO-1%[36–38]. The molar proportion of $W^{5+}$ among tungsten atoms was detected by W 4$f$ XPS spectra (Fig. S5) to be 11.5% and 17.1% for $W_{18}O_{49}$-NWs and $W_{18}O_{49}$-NWs/rGO-1%, respectively, indicating more $W^{5+}$ was present in $W_{18}O_{49}$-NWs/rGO-1%[39,40]. The light-induced increase in $W^{5+}$ was verified by the in situ XPS measurement (Fig. S6), indicating photoelectron trapping on plasmonic $W_{18}O_{49}$-NWs. In addition, their C 1$s$ XPS spectra (Fig. S7a) show that more C = O and C−O bonds are present in $W_{18}O_{49}$-NWs/rGO-

1%, indicating that $W_{18}O_{49}$-NWs and rGO may be connected via W-O-C bonds. Moreover, oxygen vacancies and surface OH groups were detected on $W_{18}O_{49}$-NWs/rGO-1% by O1$s$ XPS spectra (Fig. S7b). Thus, it was concluded that more oxygen vacancies on $W_{18}O_{49}$-NWs/rGO-1% cause strong SPR and that UV excitation can modulate SPR, which can boost hot electron generation under full-spectrum excitation.

## Photocatalytic isopropanol dehydration reaction

Abundant oxygen vacancies on nonmetallic plasmonic $W_{18}O_{49}$-NWs/rGO provide active sites for alcohol dehydration, therefore, their photocatalytic performance was tested by an isopropanol dehydration reaction (Fig. 2a). First, 0.1 mL of isopropanol was mixed with 5 mg of sample in a reaction chamber (100 mL). Under full-spectrum irradiation, $W_{18}O_{49}$-NWs/rGO-1% exhibited 109.4 mmol g⁻¹ propylene generation as a result of isopropanol dehydration in 15 min (Fig. 2b), with near 100% selectivity (Fig. S8), which is over 6-fold and 45-fold higher than those of $W_{18}O_{49}$-NWs and rGO (18.1 mmol g⁻¹ and 2.4 mmol g⁻¹), respectively. These results demonstrate that the main active sites for the isopropanol dehydration reaction were located on $W_{18}O_{49}$-NWs. The small diameter and good dispersion of $W_{18}O_{49}$-NWs on rGO can provide more active sites for the isopropanol dehydration reaction leading to greatly enhanced catalysis. The influence of their compositions on catalytic performance was investigated as shown in Fig. 2c, S9. With rGO, the isopropanol dehydration reaction was dramatically increased and $W_{18}O_{49}$-NWs/rGO-1% exhibited the optimal performance, while, is the performance decreased as the rGO content increased from 1 to 2 wt%. The propylene generation rates were calculated to be 12, 74.9, 398.2, 437.5, 321.6, and 147.6 mmol g⁻¹ h⁻¹ for rGO, $W_{18}O_{49}$-NWs, $W_{18}O_{49}$-NWs/rGO-0.5%, $W_{18}O_{49}$-NWs/rGO-1%, $W_{18}O_{49}$-NWs/rGO-1.5% and $W_{18}O_{49}$-NWs/rGO-2%, respectively, with near 100% selectivity. The isopropanol conversion efficiency was estimated to be 2.3, 14.3, 76.1, 83.6, 61.5, and 28.2% in 15 min, respectively. rGO stabilized oxygen vacancies to enhance the SPR of $W_{18}O_{49}$-NWs/rGO, promoting isopropanol dehydration for propylene. However, when the rGO content was >1 wt%, the surface of $W_{18}O_{49}$-NWs was

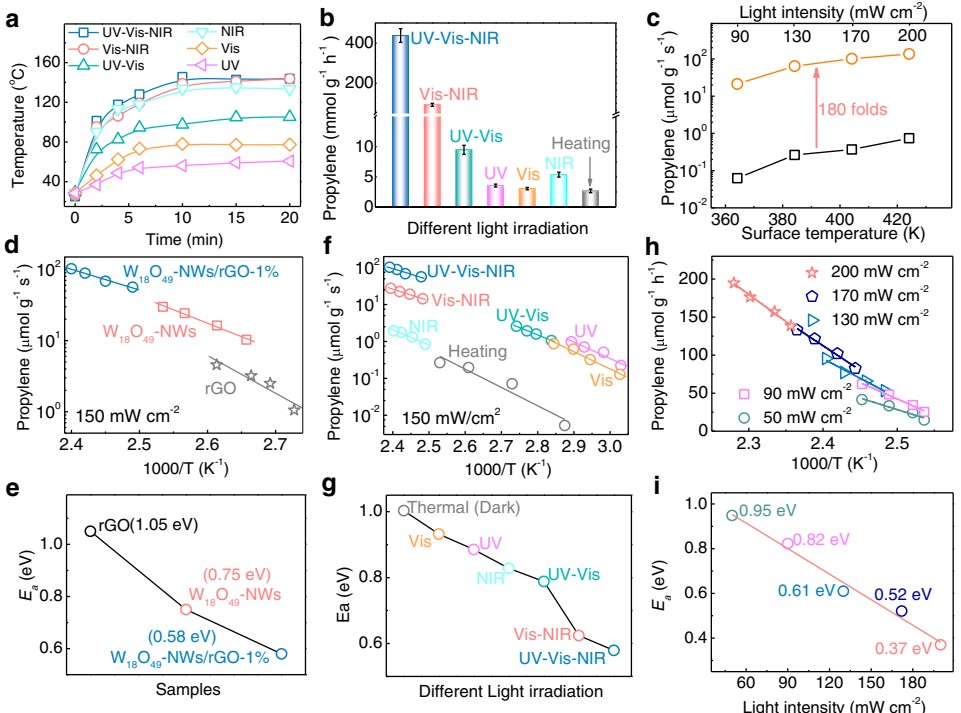

**Fig. 3 | Light-dependent reaction rate and activation barrier. a** Different light irradiation induces surface temperature variations on plasmonic $W_{18}O_{49}$-NWs/rGO-1%, and **b** their propylene generation rates from isopropanol dehydration. Bars represent mean values ± SD. **c** Photocatalytic and thermocatalytic isopropanol dehydration rates over plasmonic $W_{18}O_{49}$-NWs/rGO-1% at different surface temperatures. **d, f, h** Arrhenius plots of apparent activation barriers for isopropanol dehydration over different photocatalysts under full-spectrum light irradiation (**d**), over $W_{18}O_{49}$-NWs/rGO-1% with different light (UV, Vis, NIR, UV–Vis, Vis–NIR and full-spectrum) irradiation (**f**) and full-spectrum light irradiation with different intensities (**h**), and **e, g, i** their calculated activation barriers.

covered by rGO and the surface active site exposure was restricted for catalysis. The stability of $W_{18}O_{49}$-NWs/rGO-1% during the isopropanol dehydration reaction was investigated by 12 recycles in 3 days (Fig. 2d) and 10 h of continuous reaction (Fig. 2e), and their performance showed no substantial decrease. The TEM image and XRD patterns of $W_{18}O_{49}$-NWs/rGO-1% after photocatalysis also showed no obvious changes (Fig. S10). Consequently, plasmonic $W_{18}O_{49}$-NWs/rGO-1% with a strong SPR band and more active site exposure exhibited optimal photocatalytic performance in the isopropanol dehydration reaction.

Considering that the bioproduction of isopropanol is generally accompanied by ethanol in aqueous solution, the activity and selectivity of plasmonic $W_{18}O_{49}$-NWs/rGO on the dehydration of different alcohols were investigated, as shown in Fig. 2f. Under full-spectrum light irradiation, n-propanol dehydration over $W_{18}O_{49}$-NWs/rGO-1% exhibited 54.9 mmol g$^{-1}$ h$^{-1}$ propylene generation, much lower than that from isopropanol. Furthermore, the ethylene generation rate was only 20.6 mmol g$^{-1}$ h$^{-1}$ from ethanol dehydration and no products were detected from glycerol, verifying the preferential performance of $W_{18}O_{49}$-NWs/rGO-1% on catalytic isopropanol dehydration. In the case of 1 vol% isopropanol/water mixtures (Fig. 2g), 1.15 mmol g$^{-1}$ propylene was generated by $W_{18}O_{49}$-NWs/rGO-1% in 15 min. It exhibited a 10-fold enhancement to 11.2 mmol g$^{-1}$ from 10 vol% solution. Furthermore, 23.0 and 56.1 mmol g$^{-1}$ propylene were obtained from 20 and 50 vol% aqueous solution, which were 20-fold and 50-fold higher than that from a 1 vol% solution. The isopropanol conversion efficiency (Fig. 2i) was calculated to be 88.1, 85.5, 86.0, and 85.8 for 1, 10, 20, and 50 vol% aqueous solutions, respectively, with near 100% propylene generation. For isopropanol/ethanol mixtures (Fig. 2h), 1.15, 11.3, 23.4, and 57.1 mmol g$^{-1}$ propylene was generated from 1, 10, 20, and 50 vol% mixtures, respectively. The isopropanol conversion efficiency of these mixtures (Fig. 2i) was calculated to be 85.8%, 88.1%, 86.5%, and 89.3%, respectively. Furthermore, the selectivity of propylene generation was dramatically improved from 15% (1 vol% mixture) to 75, 87, and 94% for

the 10, 20 and 50 vol% mixtures, respectively. When isopropanol was mixed with ethanol and water at a ratio of 1:1:8, over 85% of the isopropanol (Fig. S11) was converted to propylene with 99% selectivity in 15 min. Consequently, the plasmonic $W_{18}O_{49}$-NWs/rGO-1% heterostructures exhibited preferential performance for photocatalytic isopropanol dehydration even in aqueous or ethanol mixtures.

## Contributions of nonmetallic plasmonic carriers and thermal effects in catalysis

Due to the nonradiative decay of plasmonic carriers, $W_{18}O_{49}$-NWs/rGO generated a strong thermal effect causing a surface temperature increase, which was detected by a thermal camera. UV–Vis−NIR irradiation induced the surface temperature of $W_{18}O_{49}$-NWs/rGO-1% to increase to 142.8 °C in 15 min (Fig. 3a, S12), much higher than the 95 °C of plasmonic $W_{18}O_{49}$-NWs (Fig. S13) and 77.4 °C of rGO (Fig. S14). The higher photocurrent of $W_{18}O_{49}$-NWs/rGO-1% than that of $W_{18}O_{49}$-NWs under full-spectrum

light irradiation indicates that some photoelectrons transferred to rGO (Fig. S15) underwent nonradiative decay for the photothermal effect. Those results indicate that $W_{18}O_{49}$-NWs are the dominant contributors and that rGO also partially contributes to the high photothermal effect of plasmonic $W_{18}O_{49}$-NWs/rGO-1%. The $W_{18}O_{49}$-NWs/rGO-1% heterostructure with strong SPR for plasmonic carriers and thermal effect promoted isopropanol dehydration for propylene. Different-light-irradiation-induced thermal effects on $W_{18}O_{49}$-NWs/rGO-1% were detected at 59.1, 77.2, 135.1, 104.9, 141.2, and 142.8 °C after 15 min of UV, Vis, NIR, UV–Vis, Vis–NIR, and UV−Vis−NIR irradiation, respectively, with a constant intensity of 200 mW cm$^{-2}$. Figures 3a, S16−20 show that the plasmonic thermal effect of $W_{18}O_{49}$-NWs/rGO-1% was mainly caused by NIR-light excited SPR. The effect of different light irradiation on the photocatalytic performance of $W_{18}O_{49}$-NWs/rGO-1% was tested by the isopropanol dehydration reaction. The propylene generation rates were 3.6, 3.2, 5.5, 9.4, 95.3, and

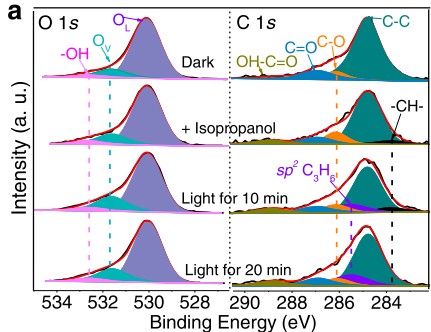
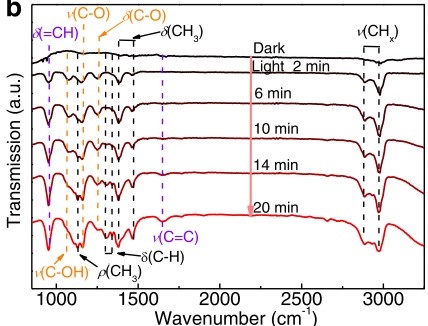

**Fig. 4 | In situ spectroscopy analysis of samples during photocatalysis. a, b** In situ O 1$s$ and C 1$s$ XPS spectra, and **c** in situ FTIR transmission spectra of W$_{18}$O$_{49}$-

NWs/rGO-1% during the isopropanol dehydration reaction under full-spectrum light irradiation.

437.5 mmol g$^{-1}$ h$^{-1}$ for UV, Vis, NIR, UV–Vis, Vis–NIR, and UV–Vis–NIR irradiation, respectively. (Fig. 3b, S21) Compared to NIR-irradiation, Vis–NIR- and UV–Vis–NIR-irradiation induced 17-fold and 79-fold enhancement in isopropanol dehydration, demonstrating that the thermal effect was not the dominant cause for the enhanced reaction. UV-irradiation generated photoelectrons for trapping to enhance the SPR of W$_{18}$O$_{49}$-NWs/rGO-1% (Fig. 1l) and facilitated hot carrier generation. Therefore, synergy between UV and Vis–NIR irradiation on W$_{18}$O$_{49}$-NWs/rGO-1% greatly accelerated the isopropanol dehydration reaction. The thermocatalytic performance of W$_{18}$O$_{49}$-NWs/rGO-1% was tested at 143 °C in the dark, and the propylene generation rate was only 2.6 mmol g$^{-1}$ h$^{-1}$. The thermocatalytic and photocatalytic isopropanol dehydration reactions were compared, as shown in Fig. 3c, and it is clear that the propylene generation rate was only improved from 0.06 to 0.74 μmol g$^{-1}$ s$^{-1}$ for thermocatalysis as the temperature increased from 91.2 to 151.1 °C. The photocatalytic propylene generation rate was improved dramatically from 21.3 to 136.6 μmol g$^{-1}$ s$^{-1}$, which is over 184-fold higher than that of thermocatalysis. These results demonstrate that the plasmonic thermal effect of W$_{18}$O$_{49}$-NWs/rGO-1% was not the key factor for the enhanced catalytic isopropanol dehydration reaction.

To quantify the contribution of the plasmonic carrier in catalytic isopropanol dehydration, the reaction rates were investigated by varying the temperature under UV–Vis–NIR-irradiation. Figure 3d shows that the production rate of propylene over plasmonic W$_{18}$O$_{49}$-NWs/rGO-1% increased as the surface temperature increased, much faster than that of plasmonic W$_{18}$O$_{49}$-NWs and rGO. By the Arrhenius equation $k = Ae^{-E_a/RT}$ (1), the apparent activation barrier ($E_a$) of the isopropanol dehydration reaction was calculated to be 1.05, 0.75 and 0.58 eV for rGO, W$_{18}$O$_{49}$-NWs and W$_{18}$O$_{49}$-NWs/rGO-1% under 150 mW cm$^{-2}$ light irradiation, respectively (Fig. 3e). The lower $E_a$ of plasmonic W$_{18}$O$_{49}$-NWs/rGO-1% was attributed to its strong SPR with more hot carrier generation. The effect of different light irradiations on $E_a$ was investigated in terms of the reaction rates, which were measured by varying the light wavelength region with a constant intensity of 150 mW cm$^{-2}$ (Fig. 3f). Their $E_a$ values were calculated to be 1.0, 0.93, 0.88, 0.83, 0.79, 0.62 and 0.58 eV for dark, Vis, UV, NIR, UV–Vis, Vis–NIR and UV–Vis–NIR irradiation, respectively (Fig. 3g). Compared to UV, Vis and NIR irradiation, full-spectrum irradiation excited the intrinsic band and SPR band of plasmonic W$_{18}$O$_{49}$-NWs/rGO simultaneously, and the continuous electron accumulation enhanced the SPR to boost hot carrier generation leading to the lowest $E_a$. The intensity of incident light determined the plasmonic carrier generation of W$_{18}$O$_{49}$-NWs/rGO, and its effect on the reaction rate was studied, as shown in Fig. 3h. The reaction rate increased as the light intensity increased from 50 to 200 mW cm$^{-2}$, and their corresponding $E_a$ values (Fig. 3i) were calculated to be 0.95, 0.82, 0.62, 0.52, and 0.37 eV, respectively, showing a linear relation as a red fitting line. The dominant contribution of monometallic plasmonic hot electrons to

enhanced catalysis was also demonstrated by ethanol and propanol dehydration reactions (Fig. S22), and their $E_a$ was reduced to 0.52 and 0.50 eV (compared to 1.22 and 1.18 eV for thermocatalysis), respectively. This evidence is sufficient to demonstrate that the $E_a$ was determined by the hot electron. Consequently, the hot electron of plasmonic W$_{18}$O$_{49}$-NWs/rGO-1% played a dominant role in boosting the isopropanol dehydration reaction by reducing $E_a$.

### Catalytic mechanism of the isopropanol dehydration reaction

The surface chemical states of W$_{18}$O$_{49}$-NWs/rGO-1% during photocatalytic isopropanol dehydration were monitored by in situ XPS measurements. The in situ O 1$s$ XPS spectra (Fig. 4a) showed that oxygen vacancies (O$_v$) (531.6 eV) and hydroxyl (−OH) (532.6 eV) increased as the light irradiation was time prolonged[27], indicating that light induced more oxygen vacancies to absorb H$_2$O for more −OH[28]. Photoelectron accumulation on plasmonic W$_{18}$O$_{49}$-NWs was verified by in situ W 4f XPS spectra (Fig. S6), and W$^{5+}$ was increased from 17.1 to 24.6% in W$_{18}$O$_{49}$-NWs/rGO-1% after 20 min of light irradiation. More information about isopropanol dehydration was obtained from the in situ C 1$s$ XPS spectra, and the enhanced peak at ~286.1 eV and new peak at ~283.1 eV assigned to C−O and −CH− bonds[41–43] verified the successful isopropanol absorption on W$_{18}$O$_{49}$-NWs/rGO-1%. Under light irradiation, the intensity of the C−O bond weakened over time, indicating the reduction of absorbed isopropanol (isopropanol*). Furthermore, a new peak at ~285.5 eV assigned to $sp^2$C appeared at 10 min and became stronger at 20 min[44], proving propylene generated from photocatalytic isopropanol dehydration. The dehydration reaction pathway of isopropanol was investigated by in situ FTIR transmission spectra of W$_{18}$O$_{49}$-NWs/rGO-1% under light irradiation. Peaks at ~1072, 1156, and 1251 cm$^{-1}$ assigned to ν(C−OH), ν(C−O), and δ(C−O) vibrations, respectively, were observed, as shown in Fig. 4b[45], which resulted from isopropanol* on plasmonic WO-NWs/rGO-1%. Upon irradiation with full-spectrum light, the intensity of those peaks decreased, suggesting the C−O bond cleavage of isopropanol. Furthermore, the intense peak at ~955 cm$^{-1}$ assigned to ν(=CH) became stronger, and a new peak assigned to ν(C=C) was observed at 1645 cm$^{-1}$, which were attributed to the intermediate of absorbed propylene (propylene*) generated from isopropanol dehydration[45]. Consequently, in situ XPS and FTIR spectra clearly show that light irradiation can promote C−O bond cleavage of isopropanol to boost the dehydration reaction for propylene.

Based on the above discussions and previous studies[23], a possible reaction mechanism of isopropanol dehydration over W$_{18}$O$_{49}$-NWs was proposed, as described in Fig. 5. Surface oxygen vacancies of plasmonic W$_{18}$O$_{49}$-NWs can absorb H$_2$O molecules to form OH moieties as Brønsted acid sites which are active in catalyzing isopropanol dehydration for propylene[12,28]. When isopropanol molecules diffuse to the surface of W$_{18}$O$_{49}$-NWs, they are coordinated to OH groups by hydroxyl accepted a proton. Then, the dehydration transition state occurs, in which the C−O bond is activated for cleavage at first state,

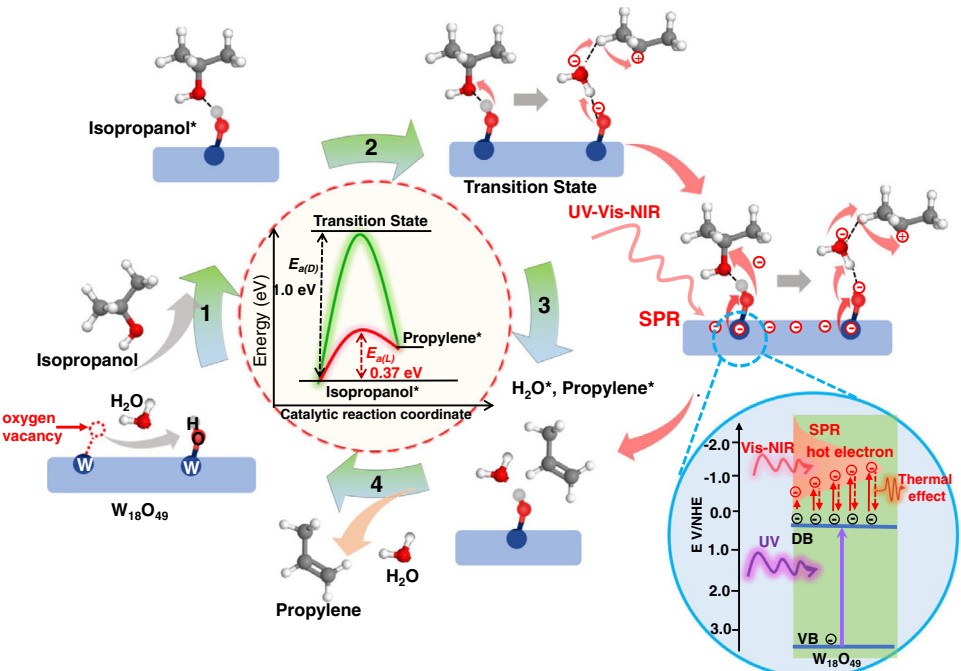

**Fig. 5 | Reaction mechanism and energy profile analysis.** Detailed reaction pathway of thermocatalytic and photocatalytic isopropanol dehydration on the surface of plasmonic $W_{18}O_{49}$, and schematic energetics of the reaction activation barrier (from step 2 to 3). Surface oxygen vacancies of $W_{18}O_{49}$-NWs can absorb $H_2O$ molecule to form OH as Brønsted acid sites. Step 1: The isopropanol molecule diffuses to the $W_{18}O_{49}$ surface to be absorbed isopropanol (Isopropanol*); Step 2: Isopropanol* is activated for the dehydration transition state with C–O bond cleavage and water molecule elimination; Step 3: Isopropanol is dehydrated to be absorbed $H_2O$ ($H_2O$*) and propylene (Propylene*); Step 4: $H_2O$* and Propylene* are released from the $W_{18}O_{49}$ surface. The blue cycle inset at right side shows full-spectrum-modulated plasmonic electron mechanism on $W_{18}O_{49}$. UV-excited electron transfer from the valence band (VB) to the defect band (DB), causing increased electron density and enhanced surface plasmon resonance (SPR), furthermore, the SPR is excited by Vis–NIR light generating hot electrons that can promote electron transfer during the isopropanol dehydration transition state.

and subsequently the incipient water molecule mediates $\beta$-elimination by accepting a proton from the $C_\beta$-carbon, which donates a different proton back to the active site of the $W_{18}O_{49}$ catalyst[23,46]. During the reaction pathways, the transition state of C-O bond breaking is the limiting step determining the dehydration reaction activation barrier as described in Fig. 5. In the case of $W_{18}O_{49}$-NWs/rGO-1%, the $E_{a(D)}$ of thermocatalysis is estimated to be 1.0 eV in the dark. With abundant oxygen vacancies, UV-excited electron transfer from the valance band (VB) to the defect band (DB) close to the conduction band of plasmonic $W_{18}O_{49}$ for accumulation facilitates the electron transfer during the transition state of isopropanol dehydration, resulting in a low $E_{a(L)}$ of 0.88 eV. The surface electrons for SPR can be excited by Vis-NIR light to generate highly active hot electrons reducing $E_{a(L)}$ to 0.62 eV. However, the fast nonradiative decay of hot electrons causes a strong thermal effect restricting hot electrons for enhanced catalysis. Under full-spectrum irradiation, the UV-excited electrons have a continuous injection from VB to DB enhancing SPR for hot electron generation, furthermore, the decay of hot electrons is restricted for long lifetimes. In this case, the electron transfer for C–O bond cleavage is greatly promoted for a lowest $E_{a(L)}$ of 0.37 eV, leading to the highest reaction rate of isopropanol dehydration. Therefore, the nonmetallic plasmonic $W_{18}O_{49}$-NWs/rGO heterostructure with full-spectrum-modulated SPR can promote hot electron generation to reduce the reaction activation barrier for enhanced photocatalysis.

## Discussion
The dominant contribution of nonmetallic plasmonic carriers in enhanced catalysis was demonstrated by constructing plasmonic $W_{18}O_{49}$-NWs/rGO heterostructures for catalytic isopropanol dehydration reactions. The rGO was combined with plasmonic $W_{18}O_{49}$-NWs to stabilize the oxygen vacancies for enhanced SPR; moreover, the UV-excited intrinsic band of $W_{18}O_{49}$-NWs induces an electron accumulation for their SPR modulation. Therefore, the intrinsic band and SPR of plasmonic $W_{18}O_{49}$-NWs/rGO heterostructures are excited simultaneously by full-spectrum irradiation to synergistically boost hot electron generation, which promotes electron transfer for C–O bond cleavage during isopropanol dehydration, resulting in the lowest activation barrier of 0.37 eV. Isopropanol dehydration to achieve nearly 100% propylene occurred with a rate of 437 mmol $g^{-1}$ $h^{-1}$, over 180-fold higher than thermocatalysis. Consequently, the flexibly modulating SPR of plasmonic semiconductors opens a new door to explore highly active plasmonic materials with enhanced hot carrier generation for photocatalysis.

## Methods
### Catalyst preparation
The $W_{18}O_{49}$-NWs/rGO composite was synthesized by using a solvothermal method. In a typical procedure, 150 mg of $WCl_6$ powder was dissolved into 30 mL of ethanol, and which was vigorously stirred to obtain a yellow suspension. Subsequently, 0.75 mL of GO solution (2 mg/mL) was added to the above solution. After vigorous magnetic stirring for 15 min, the mixture was transferred to a 50 mL Teflon-lined stainless-steel autoclave. After solvothermal treatment at 180 °C for 24 h, the sample was separated from the solution. Before further use and characterization, the sample was dried in a vacuum oven at 60 °C for 12 h. $W_{18}O_{49}$-NWs/rGO samples with different compositions were synthesized following the above experiment set using 0.375, 0.75, 1.125, and 1.50 mL of GO solution, respectively.

### Catalytic isopropanol dehydration reaction tests
A 5 mg sample was plastered on a cover glass and then placed on the bottom of the reaction chamber (100 mL). Then, the chamber was

sealed with thick quartz cover glass and degassed with pure nitrogen for 20 min. Subsequently, 0.1 mL isopropanol solution was injected into the reaction chamber and full-spectrum (UV–Vis–NIR) light (Perfectlight, PLS-SXE300D) was irradiated for the reaction. The gaseous products were detected by a gas chromatograph (Shimadzu, GC-2014A) equipped with one TCD and two flame ionization detectors (FID). Other photocatalytic reaction measurements followed the same procedure with different light irradiation which was realized by using different light-cutoff filters (UV: 200–400 nm, visible: 420–780 nm, Vis-NIR: >420 nm, NIR: >800 nm). The temperature of the reaction chamber was controlled by using a thermal oil bath. All light irradiation was maintained at a constant intensity of 200 mW/cm².

## Activation barrier calculation

The reaction activation barrier of isopropanol dehydration is calculated by the Arrhenius equation as follows:

$$lnk = lnA - \frac{E_a}{RT} \tag{1}$$

where $k$ is the reaction rate constant, $A$ is the Arrhenius constant, $E_a$ is the activation barrier, $R$ is the molar gas constant, and $T$ is the surface temperature of the catalysts. The surface temperature of the catalysts was detected by a thermal camera (FLUKE TiS65 THERMAL IMAGER).

## In situ FTIR measurement

The in situ FTIR transmission spectra were measured by an in situ diffuse reflectance Fourier transform infrared spectrometer (Bruker Tensor II FTIR NEXUS). Before the experiment, the sample was degassed for 4 h at 150 °C. Then each sample was purged with nitrogen for 1 h to blow out all the gases adsorbed on the samples. After that, a mixed gas of 0.1 mL isopropanol and water vapor was flowed into the specimen chamber for another 30 min to ensure sorption equilibrium before irradiation. The FTIR transmission spectra of the sample were recorded every 2 min.

## Catalyst characterization

The X-ray diffraction (XRD) patterns of the sample were obtained by a Rigaku Rint-2500 diffractometer with Cu K$_\alpha$ radiation at a scanning rate of 0.1° s⁻¹. The morphologies were observed by transmission electron microscopy (JEOL, 2100, operated at 100 kV) and high-resolution TEM (JEM-3000F, operated at 300 kV). The high-angle annular dark-field scanning TEM images were collected by a spherical aberration corrected transmission electronic microscope (JEM-2100 F, JEOL) equipped with double spherical aberration (Cs) correctors for both the probe-forming and image-forming objective lenses. The in situ XPS spectra were collected using an ESCALAB 250Xi+ analyzer, with Al K$\alpha$ (h$\nu$ = 1486.6 eV) as the excitation source. EPR signals were collected from a Bruker A300 spectrometer. Raman spectra were obtained from a Raman microscope (HORIBA XPLORA PLUS) with a 532 nm laser as the xcitation source. UV–vis–NIR diffuse reflectance spectra (DRS) were recorded by a UV–vis/NIR spectrophotometer (JASO V-570).

## Data availability

All the data supporting the findings of this study are available within the article and its Supplementary Information or from the corresponding authors upon reasonable request. Source data are provided with this paper.

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

## Acknowledgements

This work was supported by the National Natural Science Foundation of China (Nos. 22175076 and 51872125 for Z.L., 62035006 and 61975068 for T.G., 62204097 for D. Y., 11904133 for J.L.), the Guangdong Natural Science Funds for Distinguished Young Scholars (No. 2018B030306004, Z.L.) and the Guangdong Outstanding Scientific Innovation Foundation (No. 2019TX05X383, T.G.) and the Program of Marine Economy Development Special Fund (Six Marine Industries) under the Department of Natural Resources of Guangdong Province (No. GDNRC [2021]33, T.G.).

## Author contributions

C.L. and D.Y. synthesized the samples for characterization measurements. C.L., D.Y., J.L., L.W., T.G., and Z.L. analyzed the results. T.G. and B.L. reviewed and provided the advice on the work. Z.L. and T.G. wrote the manuscript.

## Competing interests
The authors declare no competing interests.
