## [Peer Review File · Nature Communications]

Full-spectrum nonmetallic plasmonic carriers for efficient isopropanol dehydrationReviewers' Comments:

Reviewer #1:

Remarks to the Author:

In this work, the authors proposed a nonmetallic plasmonic heterostructure by growing WO₃-x nanowires on rGO which showed strong SPR effect and highly active in photocatalytic isopropanol to produce ethylene. The author(s) thoroughly investigated the origin of the high photocatalytic performance which is attributed to plasmonic effect rather than thermal effect. Compare to conventional industrial method to produce ethylene with high energy consumption, the photocatalysis solution provided in this work more energy- and environmental-friendly. This is a novel and meaningful work which meets the requirement of Nature Communications. I would like to recommend it for publication in the journal after following concerns to be well addressed:

1. It can be seen in Fig.1i, the UV-visible-NIR DRS of WO-NWs and WO-NWs/rGO-1% are quite close, however, these two samples showed quite different photocatalytic isopropanol dehydration performance shown in Fig. 2c, can author(s) explain this phenomenon?
2. The expression in explaining stability of WO-NWs/rGO-1% in line 153 and 154, "...is investigated by 12 recycles in 3 days, and their performance show no remarkable decrease" is misleading. Actually, the reactions were tested for 60mins each day. To better clarify the stability of this sample, a continuous long-time reaction may be required as current reactions were tested within 15mins each time.
3. An energy diagram showing alignments of samples' band structure and reaction potentials is important and essential for photocatalytic reaction. Can author provide a schematic of energy diagram?
4. A blank photocatalytic test without catalysts shall be added.
5. Oxygen vacancies or surface OH group can also be identified by high resolution O 1s XPS. The author(s) shall add more discussion in the XPS part between line 116 to 122.

Reviewer #2:

Remarks to the Author:

The manuscript entitled "Full-Spectrum-Modulated Nonmetallic Plasmonic Carrier for Enhanced Catalytic Efficiency" is well written. The manuscript may be accepted after the following comments are addressed:

- 1) Why there is a reduction in IPA dehydration beyond 1% loading of RGO? Is it due to light blockage?
- 2) Are the RGO acting as hole transporter or electron transported? This need to be proved by some spectroscopic method.
- 3) The discussion on Raman for RGO is not clear. What is the ratio of D/G for GO and RGO in Raman?
- 4) The oxygen functionality in RGO has no effect on catalysis?
- 5) Can you show a better TEM image to demonstrate the composite formation?

Reviewer #3:

Remarks to the Author:

Recommendation: reject.

This manuscript reported a nonmetallic plasmonic heterostructure, WO₃-x-nanowires/reduced-graphene-oxides, as a good photocatalyst candidate for isopropanol dehydration to propylene with high product generation rate. They studied the synergetic effect of hot electron oscillation and photothermal effect in the conduction band of WO₃-x on the activity. It is an interesting topic and provides a forward-looking route for propylene production from biomass fuels. However, this manuscript has many technical problems, and I am afraid that it cannot reach the high-quality

requirements for its publication of Nature Communications. Several concerns and suggestions from the reviewer are listed as follow:

1. Novelty and significance. The application of tungsten oxide for photocatalytic alcohol dehydration has been recently reported by several papers (such as Refs. 26-28 listed in the manuscript). The methodologies for mechanism study, i.e., using different light irradiation to investigate the catalytic activity, have also been used in the above papers. Besides, the proposed reaction pathway is also quite similar with Ref. 28. These technical disadvantages may largely lower the novelty and significance of this manuscript.
2. Catalytic active sites. In the discussed mechanism in Figure 4, surface -OH species is proposed as active sites for isopropanol adsorption and the following dehydration. I have several confusions about this mechanism. 1) From O and C 1s XPS results, I do not observe OH species fingerprint on WO_{3-x} (only HO-C=O species in rGO). 2) The author declare that oxygen vacancies are active sites rather than OH species. 3) The hot electrons mainly accumulate in CB, which is mainly contributed by W d-orbitals. How do these hot electrons transfer to C-O bond via -OH species?
3. Roles of rGO. The authors demonstrate that the hot electron oscillation in conduction band of tungsten oxide. In many studies about photocatalyst heterostructures, rGO is widely used as an electron reservoir to receive the photogenerated electrons from photocatalyst terms. Does the author consider this scenario?
4. XPS results. The authors deconvoluted W 4f XPS with W⁶⁺ and W⁵⁺ species. According to the W/O atomic ratio of W₁₈O₄₉, i.e., 2.72, the relative W⁶⁺:W⁵⁺ intensity should be near 0.44:0.56. However, the observed XPS intensities in the manuscript are much different from the calculated species ratio. Besides, it is reported W₁₈O₄₉ compound contains W⁴⁺ species, which should be fully considered.
5. Catalytic performance. I am confused that different alcohol sources exhibits much different alkene production activity. According to the proposed reaction pathway, it seems alcohol source should not give rise to such activity differences. I suggest the author contribute to more discussion about this issue.
6. Product selectivity. The authors declared the near 100% selectivity for propylene production. How did the authors obtain this selectivity value? Did the author detect any other possible by-products? Besides, the word "near" should not be omitted in Abstract, Introduction and Discussion parts.
7. Photothermal effect. In Figure 3a, it is recognized the UV and visible lights can still contributes to temperature increment by themselves. So why are the observed photothermal temperatures quite comparable in the irradiation conditions of NIR, vis-NIR and UV-vis-NIR?
8. Sample nomenclatures. "WO_{3-x}" and "WO-NWs" may cause misunderstanding and it may be better to be replaced by a more accurate term because tungsten oxides contain many nonstoichiometric compounds. According to the characterizations, nonstoichiometric W₁₈O₄₉ is confirmed.

Manuscript ID: NCOMMS-22-30343

Title: Full-Spectrum-Modulated Nonmetallic Plasmonic Carrier for Enhanced Catalytic Efficiency

Point-by-Point Response to Reviewers

Reviewer: 1

Comments:

In this work, the authors proposed a nonmetallic plasmonic heterostructure by growing WO_{3-x} nanowires on rGO which showed strong SPR effect and highly active in photocatalytic isopropanol to produce ethylene. The author(s) thoroughly investigated the origin of the high photocatalytic performance which is attributed to plasmonic effect rather than thermal effect. Compare to conventional industrial method to produce ethylene with high energy consumption, the photocatalysis solution provided in this work more energy- and environmental-friendly. This is a novel and meaningful work which meets the requirement of Nature Communications. I would like to recommend it for publication in the journal after following concerns to be well addressed:

1. It can be seen in Fig. 1i, the UV-visible-NIR DRS of WO-NWs and WO-NWs/rGO-1% are quite close, however, these two samples showed quite different photocatalytic isopropanol dehydration performance shown in Fig. 2c, can author(s) explain this phenomenon?

Response: Thank you for reviewer's comments. Actually, the photocatalytic efficiency is determined by both light absorption and surface active sites of photocatalysts. As shown in the revised Fig. 1b, the synthesized $\text{W}_{18}\text{O}_{49}$ nanowires ($\text{W}_{18}\text{O}_{49}$ -NWs) without rGO have larger dimeters (10-14 nm) than $\text{W}_{18}\text{O}_{49}$ -NWs (5 nm) loaded on rGO. It means that rGO can act as substrate to control the growth of $\text{W}_{18}\text{O}_{49}$ -NWs with small dimeters and well dispersion. Therefore, the thin $\text{W}_{18}\text{O}_{49}$ -NWs loaded on rGO have more active site for isopropanol dehydration reaction. In additions, rGO can stabilize their oxygen vacancies from their EPR (Fig. 1m) and XPS (Fig. S5-7) spectra, and which can enhance SPR band of $\text{W}_{18}\text{O}_{49}$ -NWs/rGO-1% for more hot electron generation and high photothermal effect (Fig. S12-14) to boost catalysis. As rGO is over 1%, the light absorption of $\text{W}_{18}\text{O}_{49}$ -NWs/rGO-1.5% and WO-NWs/rGO-2% (Fig. S4) have much enhancement, while, their $\text{W}_{18}\text{O}_{49}$ -NWs surface is covered by rGO and the active site exposure is limited, resulting in a decreased catalytic efficiency (Fig. 2c).

Modifications: We added the sentence "The thin dimeter and well dispersion of $\text{W}_{18}\text{O}_{49}$ -NWs on rGO can provide more active sites for isopropanol dehydration reaction leading to greatly enhanced catalysis." at p5 of the revised manuscript.

2. The expression in explaining stability of WO-NWs/rGO-1% in line 153 and 154," is investigated by 12 recycles in 3 days, and their performance show no remarkable decrease" is misleading. Actually, the reactions were texted for 60mins each day. To better clarify the stability of this sample, a continuous long-time reaction may be required as current reactions were texted within 15mins each time.

Response: Thank you for reviewer's suggestion. We measured the long continuous reaction in 10 hours. As shown in Fig. 2e, the dehydration generation has a linear increase as time prolongs.

Fig. 2e. Long time continuous isopropanol dehydration reaction for propylene (10 hours).

Modifications: We revised the sentence “The stability of $W_{18}O_{49}$ -NWs/rGO-1% during isopropanol dehydration reaction is investigated by 12 recycles in 3 days, and their performance show no remarkable decrease (Fig. 2d).” to “The stability of $W_{18}O_{49}$ -NWs/rGO-1% during isopropanol dehydration reaction is investigated by 12 recycles in 3 days (Fig. 2d) and 10 hours continuous reaction (Fig. 2e), and their performance show no remarkable decrease.” at p5 of the revised manuscript.

Fig. 2: Photocatalytic isopropanol dehydration to propylene.

3. An energy diagram showing alignments of samples' band structure and reaction potentials is important and essential for photocatalytic reaction. Can author provide a schematic of energy diagram?

Response: The band structure potentials of plasmonic $W_{18}O_{49}$ and detailed isopropanol dehydration reaction activation barriers were added in the revised Fig. 5.

Fig. 5: Reaction mechanism and energy profile analysis.

4. A blank photocatalytic test without catalysts shall be added.

Response: A blank photocatalytic test without catalysts have been given in the revised Fig. 2b.

Fig. 2b Propylene generation from isopropanol dehydration over rGO, $W_{18}O_{49}$ -NWs, $W_{18}O_{49}$ -NWs/rGO-1% and without catalysts under full-spectrum-irradiation.

5. Oxygen vacancies or surface OH group can also be identified by high resolution O 1s XPS. The author(s) shall add more discussion in the XPS part between line 116 to 122.

Response: Thank you for reviewer's suggestion. We measured the high resolution O1s XPS spectra of $W_{18}O_{49}$ -NWs/rGO-1% as shown in Fig. S7b.

Modifications: We added the sentence "Moreover, oxygen vacancy and surface OH group are detected on $W_{18}O_{49}$ -NWs/rGO-1% by O1s XPS spectra (Fig. S7b)." at p5 of the revised manuscript.

Fig. S7b: O 1s XPS spectra of $W_{18}O_{49}$ -NWs and $W_{18}O_{49}$ -NWs/rGO-1%. O_L: Oxygen in crystal lattices. O_v: Oxygen vacancies.

Reviewer: 2

Comments:

The manuscript entitled “Full-Spectrum-Modulated Nonmetallic Plasmonic Carrier for Enhanced Catalytic Efficiency” is well written. The manuscript may be accepted after the following comments are addressed:

1) Why there is a reduction in IPA dehydration beyond 1% loading of RGO? Is it due to light blockage?

Response: Thank you for reviewer’s comments. The photocatalytic results in **Fig. 2b** show the main active sites for isopropanol dehydration located on plasmonic $W_{18}O_{49}$ -NWs. As rGO is 1%, $W_{18}O_{49}$ -NWs with thin diameter and well dispersion (**Fig. 1c-j**) can provide more active sites for isopropanol dehydration reaction, exhibiting optimal photocatalytic efficiency. However, as rGO is over 1%, the surface of $W_{18}O_{49}$ -NWs is covered by rGO (**Fig. S1**) which can stabilize more oxygen vacancies for their stronger SPR (**Fig. S4**) but limits the active site exposure for reaction, resulting in a decreased catalysis.

2) Are the RGO acting as hole transporter or electron transported? This need to be proved by some spectroscopic method.

Response: Due to the abundant oxygen vacancies of $W_{18}O_{49}$ -NWs, it is difficult to measure the PL spectra of $W_{18}O_{49}$ -NWs/rGO heterostructures for investigation of the carrier separation and transfer process. While, we measured the photocurrent of $W_{18}O_{49}$ -NWs/rGO heterostructures as shown in **Fig. S15**. It is clear that the $W_{18}O_{49}$ -NWs/rGO exhibits a remarkable enhanced photocurrent compared to $W_{18}O_{49}$ -NWs under full-spectrum light irradiation (Fig. S15a), indicating the possible of photoelectron transfer from $W_{18}O_{49}$ -NWs to rGO. Different-light-induced photocurrents (Fig. S15b) show that the UV-excited interband transition can promote the electron transfer from plasmonic $W_{18}O_{49}$ -NWs to rGO, then, a nonradiative decay generates photothermal effect. It is why UV-irradiation can cause temperature of $W_{18}O_{49}$ -NWs/rGO increased to 59 °C.

Modifications: We added the sentence “Higher photocurrent of $W_{18}O_{49}$ -NWs/rGO-1% than that of WO-NSs under full-spectrum light irradiation indicates the part photoelectrons transferred to rGO (**Fig. S15**) having a nonradiative decay for photothermal effect.” at p7 of revised manuscript.

Fig. S15. Photocurrents of $W_{18}O_{49}$ -NWs/rGO-1% and WO-NWs under full-spectrum light irradiation.

3) The discussion on Raman for RGO is not clear. What is the ratio of D/G for GO and RGO in Raman?

Response: Actually, we did more discussion about Raman spectra of rGO in the supporting information as “Chemical structure of rGO layers in $W_{18}O_{49}$ -NWs/rGO-1% is analyzed by the Raman spectra (Fig. S4), and in which two peaks are observed at 1580 and 1350 cm^{-1} corresponding to the G and D bands of rGO.³ The G band is assigned to the vibration of sp^2 -bonds, while, the D band is correlated with defects derived from grain boundaries and vacancies.⁴ The I_D/I_G ratio is calculated to 1.00 and 1.15 for rGO and $W_{18}O_{49}$ -NWs/rGO-1%, respectively, implying more vacancies in the heterostructure.⁵” We also added Raman spectra of GO as shown in the revised Fig. S3, and their I_D/I_G ratio is 0.86, lower than 1.00 of rGO.

Modifications: We revised the sentence “X-ray diffraction patterns (Fig. S3) imply the crystal structure of nanowires assigned to $W_{18}O_{49}$ and the high intensity ratio of D-band/G-band^{32,33} in Raman spectra (Fig. S4) indicate more oxygen vacancies in the heterostructure.³⁴” to “X-ray diffraction patterns (Fig. S2) imply the crystal structure of nanowires assigned to $W_{18}O_{49}$. Their Raman spectra (Fig. S3) show two peaks located at 1580 and 1350 cm^{-1} corresponding to the G and D bands of rGO.³² The G band is assigned to the vibration of sp^2 -bonds, while, the D band is correlated with defects derived from grain boundaries and vacancies.³³ The I_D/I_G ratio is calculated to 0.86, 1.00 and 1.15 for GO, rGO and $W_{18}O_{49}$ -NWs/rGO-1%, respectively, implying more oxygen vacancies in the heterostructure.³⁴” at p3 of the revised manuscript.

Fig. S3 Raman spectra of WO-NWs/rGO-1%, $W_{18}O_{49}$ -NWs, rGO and GO.

4) The oxygen functionality in RGO has no effect on catalysis?

Response: In this work, the active sites for photocatalytic isopropanol dehydration reaction are main located on plasmonic $W_{18}O_{49}$ -NWs, while, rGO have low activity for reaction (Fig. 2b). So, we did not investigate the effect of oxygen functionality in rGO on catalysis. Moreover, $W_{18}O_{49}$ -NWs/rGO heterostructures were synthesized by one-step solvothermal treatment, and it is difficult to control the amount of oxygen functionality on surface of rGO. However, from their C

1s XPS spectra (Fig. S7a), more C-O bonds are detected in $W_{18}O_{49}$ -NWs/rGO-1%, indicating that the connection between $W_{18}O_{49}$ -NWs and rGO may through W-O-C bonds. Therefore, the oxygen functionality in rGO can facilitate the well dispersion of $W_{18}O_{49}$ -NWs, which are important for the activity of plasmonic $W_{18}O_{49}$ -NWs.

5) Can you show a better TEM image to demonstrate the composite formation?

Response: Thank you for reviewer's suggestion. The clear TEM images of rGO, $W_{18}O_{49}$ -NWs and $W_{18}O_{49}$ -NWs/rGO-1% were added in the revised Fig. 1.

Fig. 1: Characterizations of catalysts with various analysis techniques. a-b TEM of rGO and $W_{18}O_{49}$ -NWs. **c, d** TEM, **e** HRTEM, **f** HAAD-STEM and **g-j** EDS elements (W, O, C) mapping images of $W_{18}O_{49}$ -NWs/rGO-1% heterostructures. **k** UV-visible-NIR DRS of rGO, $W_{18}O_{49}$ -NWs and $W_{18}O_{49}$ -NWs/rGO-1%, respectively. **l** UV-irradiation-induced DRS varies of $W_{18}O_{49}$ -NWs/rGO-1%. **m** EPR spectra of $W_{18}O_{49}$ -NWs and $W_{18}O_{49}$ -NWs/rGO-1%.

Reviewer: 3

Comments:

This manuscript reported a nonmetallic plasmonic heterostructure, WO_{3-x} -nanowires/reduced-graphene-oxides, as a good photocatalyst candidate for isopropanol dehydration to propylene with high product generation rate. They studied the synergetic effect of hot electron oscillation and photothermal effect in the conduction band of WO_{3-x} on the activity. It is an interesting topic and provides a forward-looking route for propylene production from biomass fuels. However, this manuscript has many technical problems, and I am afraid that it cannot reach the high-quality requirements for its publication of Nature Communications. Several concerns and suggestions from the reviewer are listed as follow:

Response: Thank you for reviewer's comments. Actually, several literatures (Ref. 26-28) have reported the synergetic effect of plasmonic hot carriers and photothermal effect on enhanced ethanol dehydration. However, quantifying the detailed contributions of nonmetallic plasmonic hot carriers and photothermal effect on catalysis is very critical for exploring the high-active nonmetallic plasmonic photocatalysts. So, we evaluated the contributions of nonmetallic plasmonic hot electron and photothermal effect on enhanced catalysis by calculating the reaction activation barriers (E_a) of isopropanol dehydration during photocatalysis and thermocatalysis, and the nonmetallic plasmonic hot electron was demonstrated to make dominant contribution on enhanced reaction via reducing the E_a . A synergy between UV-excited interband transition and Vis-NIR-induced SPR excitation on plasmonic $\text{W}_{18}\text{O}_{49}$ -NWs/rGO heterostructures was verified to boost hot electron generation which can promote electron transfer for C-O bond cleavage during transition state of isopropanol dehydration, leading to the low E_a down to 0.37 eV (compare to 1.0 eV of thermocatalysis) and fast propylene generation rate up to 437.5 mmol g⁻¹ h⁻¹ (near 100% selectivity). So, our work clears the dominant contribution of nonmetallic plasmonic hot carriers on enhanced catalysis, and will help guide the design of better catalytic chemistries.

The more explanations on novelty of our work will be given in the detailed response to each comment as below, and we hope our response and modifications can meet the approval of reviewer.

1. Novelty and significance. The application of tungsten oxide for photocatalytic alcohol dehydration has been recently reported by several papers (such as Refs. 26-28 listed in the manuscript). The methodologies for mechanism study, i.e., using different light irradiation to investigate the catalytic activity, have also been used in the above papers. Besides, the proposed reaction pathway is also quite similar with Ref. 28. These technical disadvantages may largely lower the novelty and significance of this manuscript.

Response: The tungsten oxide for photocatalytic alcohol dehydration was reported by our group at 2020 (Ref. 26, *Appl. Catal. B-Environ.* 264, 118517 (2020)), then by Ma and Ye et al (Ref. 28, *Adv. Funct. Mater.* 2110026 (2021)), and recently by Xiong et al (Ref. 27, *JACS Au* 2, 1160-1168 (2022)).

1. In Ref.26, we investigated the photocatalytic ethanol dehydration over plasmonic WO_{3-x} as catalysts, and concluded that the plasmonic hot electrons and photothermal effect synergistically

contribute to the superior photocatalytic performance of WO_{3-x} nanowires in ethanol dehydration.

2. In Ref.27, Xiong et al studied the detailed reaction pathway of ethanol dehydration over $\text{W}_{18}\text{O}_{49}$ as photocatalysts by using *in situ* XPS spectra measurement, and concluded that the energetically “hot” electrons excited by intraband transition can effectively modulate the charge dynamics, enhancing ethanol dehydration process, and a concomitant photothermal effect supplies a local elevated reaction temperature to further promote the process of alcohol dehydration.

3. In Ref.28, Ma and Ye et al synthesized WO_{3-x}/C composites and concluded that WO_{3-x} induces the LSPR effect and solid acid center for adsorption and activation capacity of $\text{C}_2\text{H}_5\text{OH}$ and the C coating further enhances the photo-thermal synergy, ensuring the high reaction temperature and hot carrier transmission simultaneously.

Figure. **a** Different light irradiation induces propylene generation rates from isopropanol dehydration over plasmonic $\text{W}_{18}\text{O}_{49}$ -NWs/rGO-1%. **b** Photocatalytic and thermocatalytic isopropanol dehydration rates at different surface temperatures. **c** Isopropanol dehydration reaction activation barriers (E_a) under different light irradiation and different light intensities.

In above Ref.26-28, the plasmonic hot carrier and thermal effect were considered to play a synergy role on boosting photocatalytic ethanol dehydration. However, their detailed contributions are still unclear. To quantify the contributions of nonmetallic plasmonic hot carriers and thermal effect on catalysis, we proposed a new strategy for flexibly modulating the hot electron using a nonmetallic plasmonic heterostructure (named $\text{W}_{18}\text{O}_{49}$ -nanowires/reduced-graphene-oxides). We evaluated the contributions of different light irradiation and thermal effect on isopropanol dehydration as shown in above figure **a** and **b**, and it is clear that the great

enhancement on propylene generation is caused by photocatalysis. Photocatalytic isopropanol dehydration reaction rate was greatly enhanced to 180 folds compared to thermocatalysis, and up to 437 mmol g⁻¹ h⁻¹ propylene generation rate is achieved over W₁₈O₄₉-NWs/rGO-1% under 200 mW cm⁻² full-spectrum light irradiation. Detailed mechanism of light-enhanced isopropanol dehydration was investigated by calculating their reaction activation barriers (E_a) through Arrhenius equation $k = Ae^{-E_a/RT}$ under different light irradiations and different light intensities. From above figure **c**, it is clear that the isopropanol dehydration E_a is reduced to 0.88 and 0.62 eV by UV-excitation and Vis-NIR-excitation, respectively, lower than 1.0 eV of thermocatalysis. Much interesting is that full-spectrum irradiation excited interband transition and SPR band simultaneously have a synergy on reducing E_a down to 0.37 eV. During thermocatalytic isopropanol dehydration, electron transfer for C-O band cleavage of transition state is the key step to determine the reaction activation barrier, and the nonmetallic hot electron generated on plasmonic W₁₈O₄₉-NWs was considered to promote electron transfer for C-O band cleavage during isopropanol dehydration transition state, reducing the reaction activation barrier for enhanced propylene generation up to 437 mmol g⁻¹ h⁻¹.

The detailed differences between our manuscript and Ref. 26-28 are shown in Table 1.

Table 1. Comparisons between our manuscript and Ref. 26-28

	Ref. 26	Ref. 27	Ref. 28	This work
Sample	WO _{3-x}	W ₁₈ O ₄₉	WO _{3-x} @C	W ₁₈ O ₄₉ /rGO
Alcohol	Ethanol	Ethanol	Ethanol	Isopropanol
Products	Ethylene	Ethylene	Ethylene	Propylene
Light intensity (W/cm ²)	0.2	1.67	1.69	0.2
Temperature (°C)	80	260	230.5	142.8
Selectivity (%)	94.9	99.9	98	99.96
Generation rates (mmol/g/h)	16.9	275.5	2.8 (from 10% ethanol)	437.5
Thermocatalytic E_a (eV)	—	—	—	1.0 eV
Photocatalytic E_a (eV)	—	—	—	0.37 eV

E_a : Dehydration reaction activation barrier

2. Catalytic active sites. In the discussed mechanism in Figure 4, surface -OH species is proposed as active sites for isopropanol adsorption and the following dehydration. I have several confusions about this mechanism. 1) Form O and C 1s XPS results, I do not observe OH species fingerprint on WO_{3-x} (only HO-C=O species in rGO). 2) The author declare that oxygen vacancies are active sites rather than OH species. 3) The hot electrons mainly accumulate in CB, which is mainly contributed by W d-orbitals. How do these hot electrons transfer to C-O bond via -OH species?

Response: We measured the high-resolution C and O 1s XPS spectra as shown in the revised Fig. S7. Oxygen vacancies and little OH are detected in O 1s XPS spectra of $W_{18}O_{49}$ -NWs/rGO-1%. Oxygen vacancies on the surface of plasmonic WO-NWs will absorb H_2O molecules to generate OH species as Brönsted acid sites for isopropanol dehydration reaction. Photoelectrons excited by interband transition can transfer from valance band (VB) to defect band (DB) (closed to conduction band) for accumulations, leading to surface plasmon resonance (SPR). Then, the electrons on DB can occur oscillation with incident light generating hot electron at high energy state. In this case, the hot electron easily transfers to OH group, promoting the proton transfer and accelerating C-O bond cleavage during isopropanol dehydration.

In order to investigate the reaction pathway of isopropanol dehydration on surface of plasmonic WO-NWs/rGO-1%, the *in situ* XPS and FTIR transmission spectra measurements were carried as shown in Fig. 4a-c of the revised manuscript. Those results clearly showed that light irradiation can promote C-O cleavage of absorbed isopropanol (*isopropanol) to generate propylene.

Modifications: We added the sentences “Moreover, oxygen vacancy and surface OH group are detected on $W_{18}O_{49}$ -NWs/rGO-1% by O1s XPS spectra (Fig. S7b).” and “Surface oxygen vacancies of plasmonic $W_{18}O_{49}$ -NWs can absorb H_2O molecule to be OH as Brönsted acid sites which are active to catalyze isopropanol dehydration for propylene.^{12,28}” at p5 and p10 of the revised manuscript, respectively.

We added more discussions at p9 of the revised manuscript. “Surface chemical states of $W_{18}O_{49}$ -NWs/rGO-1% during photocatalytic isopropanol dehydration was monitored by *in situ* XPS measurements. *In situ* O 1s XPS spectra (Fig. 4a) showed that oxygen vacancies (O_v) (531.6 eV) and hydroxyl (-OH) (532.6 eV) are increased as light irradiation time prolongs,²⁷ indicating light-induced more oxygen vacancies to absorb H_2O for more -OH.²⁸ Photoelectron accumulation on plasmonic $W_{18}O_{49}$ -NWs was verified by *in situ* W 4f XPS spectra (Fig. S6), and W^{5+} was increased from 17.1% to 24.6% in $W_{18}O_{49}$ -NWs/rGO-1% after 20 min light irradiation. More information of isopropanol dehydration was observed from *in situ* C 1s XPS spectra, and the enhanced peak around 286.1 eV and new peak around 283.1 eV assigned to C-O and -CH-bonds⁴¹⁻⁴³ verified the successful isopropanol absorption on $W_{18}O_{49}$ -NWs/rGO-1%. Under light irradiation, the intensity of C-O bond became weak as time prolongs meaning the reduce of absorbed isopropanol (isopropanol*). Meanwhile, a new peak around 285.5 eV assigned to sp^2 C appeared at 10 min and became stronger at 20 min,⁴⁴ proving propylene generated from photocatalytic isopropanol dehydration. The depth understanding on dehydration reaction pathway of isopropanol was investigated by *in situ* FTIR transmission spectra of $W_{18}O_{49}$ -NWs/rGO-1% under light irradiation. The peaks around 1072, 1156 and 1251 cm^{-1} assigned to $\nu(C-OH)$, $\nu(C-O)$ and $\delta(C-O)$ vibration were observed in Fig. 4b,⁴⁵ respectively, which were resulted from isopropanol* on plasmonic WO-NWs/rGO-1%. As full-spectrum light is irradiated, the intensity of those peaks became weak, suggesting the C-O bond cleavage of isopropanol. Meanwhile, the remarkable peak around 955 cm^{-1} assigned to $\nu(=CH)$ became stronger, and a new peak assigned to $\nu(C=C)$ was observed at 1645 cm^{-1} , suggesting the intermediate of absorbed propylene (propylene*) generated from isopropanol dehydration.⁴⁵ Consequently, *in situ* XPS and FTIR spectra show clearly that light irradiation can promote C-O bond cleavage of isopropanol to boost dehydration reaction for propylene.”

We also added Fig. 4 and modified Fig. S7 and Fig. 5 in the revised manuscript and supporting information. Please see the following Figures.

Fig. 4: *In situ* spectroscopy analysis on samples during photocatalysis. **a, b** *In situ* O 1s and C 1s XPS spectra, and **c** *in situ* FTIR transmission spectra of $\text{W}_{18}\text{O}_{49}$ -NWs/rGO-1% during isopropanol dehydration reaction under full-spectrum light irradiation.

Fig. 5: Reaction mechanism and energy profile analysis.

Fig. S7 XPS spectra of C 1s (a) and O 2p (b) in W₁₈O₄₉-NWs, rGO and W₁₈O₄₉-NWs/rGO-1%, respectively.

3. Roles of rGO. The authors demonstrate that the hot electron oscillation in conduction band of tungsten oxide. In many studies about photocatalyst heterostructures, rGO is widely used as an electron reservoir to receive the photogenerated electrons from photocatalyst terms. Does the author consider this scenario?

Response: To show the carrier separation and transfer process in heterostructures, we measured the photocurrent of W₁₈O₄₉-NWs/rGO heterostructures (**Fig. S15a**). It is clear that the W₁₈O₄₉-NWs/rGO exhibits a remarkable enhanced photocurrent compared to W₁₈O₄₉-NWs under full-spectrum light irradiation, indicating the possible of photoelectron transfer from W₁₈O₄₉-NWs to rGO. In the case of catalysis, the electrons transferred to rGO will have nonradiative decay to generate photothermal effect. So, compared to plasmonic W₁₈O₄₉-NWs, W₁₈O₄₉-NWs/rGO exhibits higher surface temperature (**Fig. 3a**) during photocatalysis. Different light irradiations induced photocurrents were also measured in **Fig. 15b**. Photocurrent of NIR irradiation is weaker than other light irradiations, indicating the fast nonradiative decay of hot electrons for thermal effect. UV and UV-vis irradiation excited interband transition for photoelectron at high energy state can transfer to rGO for higher photocurrents, while, in catalysis process, the electrons transferred to rGO have a nonradiative decay for thermal effects. So, rGO layers can promote the part electron transfer from W₁₈O₄₉-NWs to rGO and finally resulting in a photothermal effect.

Modifications: We added the sentence “Higher photocurrent of W₁₈O₄₉-NWs/rGO-1% than that of W₁₈O₄₉-NWs under full-spectrum light irradiation indicates the part photoelectrons transferred to rGO (**Fig. S15**) having a nonradiative decay for photothermal effect.” at p7 of the revised manuscript.

Fig. S15. a) Photocurrents of $W_{18}O_{49}$ -NWs/rGO-1% and $W_{18}O_{49}$ -NWs under full-spectrum light irradiation. b) Different light irradiations induce photocurrents of $W_{18}O_{49}$ -NWs/rGO-1%.

4. XPS results. The authors deconvoluted W 4f XPS with W^{6+} and W^{5+} species. According to the W/O atomic ratio of $W_{18}O_{49}$, i.e., 2.72, the relative W^{6+} : W^{5+} intensity should be near 0.44:0.56. However, the observed XPS intensities in the manuscript are much different from the calculated species ratio. Besides, it is reported $W_{18}O_{49}$ compound contains W^{4+} species, which should be fully considered.

Response: We measured the high-resolution W 4f XPS spectra of $W_{18}O_{49}$ -NWs and $W_{18}O_{49}$ -NWs/rGO-1% as shown in the revised Fig. S5. The molar ratio of W^{5+} in tungsten atoms is calculated to 11.5% and 17.1% for $W_{18}O_{49}$ -NWs and $W_{18}O_{49}$ -NWs/rGO-1%, respectively, which is lower than the theoretical 56% of $W_{18}O_{49}$. Many literatures also reported the lower ratio of W^{5+} in $W_{18}O_{49}$, (*Adv. Mater.* 34, 2109330 (2022), *Adv. Funct. Mater.* 32, 220363 (2022), *Appl. Catal. B-Environ.* 254, 351–359 (2019)), and the possible reason was that the oxygen vacancies of $W_{18}O_{49}$ absorb the oxygen during drying treatment. The ratio of W^{5+} can be modulated in a broad region, from *in situ* W 4f XPS measurement (Fig. S6), it can be increased from 17.1% to 24.6% by 20 min light irradiation. No remarkable signal of W^{4+} was detected in W 4f XPS spectra of $W_{18}O_{49}$ -NWs/rGO-1%.

Modifications: We revised the sentence “It is furtherly confirmed by more W^{5+} detected in plasmonic heterostructure from W 4f XPS spectra (Fig. S5).^{39,40}” to “The molar ratio of W^{5+} in tungsten atoms is detected by W 4f XPS spectra (Fig. S5) to be 11.5% and 17.1% for $W_{18}O_{49}$ -NWs and $W_{18}O_{49}$ -NWs/rGO-1%, respectively, indicating more W^{5+} existed in $W_{18}O_{49}$ -NWs/rGO-1%. Light-induced increase of W^{5+} is verified by the *in situ* XPS measurement (Fig. S6), indicating the photoelectron trapping on plasmonic $W_{18}O_{49}$ -NWs.” at p4-5 of the revised manuscript.

We also follow the reviewer’s suggestion to revise the “ WO_{3-x} ” and “WO-NWs” to “ $W_{18}O_{49}$ ” and “ $W_{18}O_{49}$ -NWs” in the whole manuscript.

Fig. S5 W 4f XPS spectra of $W_{18}O_{49}$ -NWs and $W_{18}O_{49}$ -NWs/rGO-1%

Fig. S6 In-situ W 4f XPS spectra of $W_{18}O_{49}$ -NWs/rGO-1% under full-spectrum light irradiation.

5. Catalytic performance. I am confused that different alcohol sources exhibits much different alkene production activity. According to the proposed reaction pathway, it seems alcohol source should not give rise to such activity differences. I suggest the author contribute to more discussion about this issue.

Response: For different alcohols, their dehydration reaction barriers (E_a) have huge differences resulting in different reaction rates. In order to identify their accurate E_a , we measured the propanol and ethanol dehydration rates over $W_{18}O_{49}$ -NWs/rGO as catalysts under full-spectrum light irradiation by varying reaction temperatures and their E_a are calculated by Arrhenius equation $k = Ae^{-E_a/RT}$. The results are shown in Fig. S22, it is clear that full-spectrum-irradiation on $W_{18}O_{49}$ -NW/rGO-1% can greatly reduce the ethanol dehydration E_a down to 0.52 eV (compare to 1.22 eV of thermocatalysis), meanwhile, the propanol dehydration E_a is reduced down to 0.50 eV (compare to 1.18 eV of thermocatalysis). So, the dominant contribution of nonmetallic plasmonic hot electron on enhanced catalysis were also concluded by ethanol and propanol dehydration reactions. Herein, isopropanol dehydration reaction is used as one model to demonstrate the dominant contribution of nonmetallic plasmonic hot electron on enhanced catalysis.

Modifications: We added the sentence “The dominant contribution of monometallic plasmonic hot electron on enhanced catalysis is also demonstrated by ethanol and propanol dehydration reaction (Fig. S22), and their E_a is reduced down to 0.52 and 0.50 eV (compare to 1.22 and 1.18 eV of thermocatalysis), respectively.” at p9 of the revised manuscript.

Fig. S22. a Arrhenius plots of apparent activation barriers for ethanol and propanol dehydration over $W_{18}O_{49}$ -NW/rGO-1% under dark and full-spectrum light irradiation. b Thermocatalytic and photocatalytic ethanol and propanol dehydration rates, and their calculated activation barriers (E_a).

6. Product selectivity. The authors declared the near 100% selectivity for propylene production. How did the authors obtain this selectivity value? Did the author detect any other possible by-products? Besides, the word “near” should not be omitted in Abstract, Introduction and Discussion parts.

Response: The detailed products analysis was given in Fig. S8. Besides propylene, other products are detected with very small amount of CO, CH_4 and $\text{CH}_3\text{CH}_2\text{CH}_3$. The calculated selectivity of propylene is 99.96%. So, we use near 100% selectivity for propylene generation.

Modifications: We revised the “100% selectivity” to “near 100% selectivity” in the whole manuscript. We revised the sentence “Under full-spectrum irradiation, $W_{18}O_{49}$ -NWs/rGO-1% exhibits 109.4 mmol g⁻¹ propylene generation from isopropanol dehydration in 15 min (Fig. 2b), with near 100 % selectivity,” to “Under full-spectrum irradiation, $W_{18}O_{49}$ -NWs/rGO-1% exhibits 109.4 mmol g⁻¹ propylene generation from isopropanol dehydration in 15 min (Fig. 2b), with near 100 % selectivity (Fig. S8),” at p5 of the revised manuscript.

Fig. S8 Generation rate (a) and selectivity (b) of products from isopropanol dehydration reaction.

7. Photothermal effect. In Figure 3a, it is recognized the UV and visible lights can still contribute to temperature increment by themselves. So why are the observed photothermal temperatures quite comparable in the irradiation conditions of NIR, vis-NIR and UV-vis-NIR?

Response: In our manuscript, in order to exclude the influence of light intensity on thermal effect of $W_{18}O_{49}$ -NWs/rGO, we kept UV, Visible, NIR, Vis-NIR and UV-Vis-NIR light with a constant intensity of 200 mW cm⁻² for surface temperature measurement. From Fig. 1l, it is clear that UV-excited interband transition of $W_{18}O_{49}$ -NWs can generate electron accumulation on defect band, and part electrons can transfer to rGO which are verified by their increased photocurrent (Fig. S15) and have a nonradiative decay for thermal effect, resulting in surface temperature increase to 59.8 °C in 15 min. For visible light irradiation, SPR of WO-NWs has weak absorption, while, the excited hot electrons have fast nonradiative decay to generate thermal effect and their surface temperature is increased to 77.2 °C in 15 min. With the strong SPR absorption in NIR light region, the NIR-irradiation can excite SPR to generate more low energy hot electron for fast nonradiative decay generating strong thermal effect, therefore, NIR-irradiation induces the surface temperature increased to 135.1 °C in 15 min. For Vis-NIR and UV-vis-NIR light with the constant 200 mW cm⁻², NIR part has the main contribution and Vis/UV-Vis parts also have contribution on thermal effect, so, their temperature is increased to 142.8 °C in 15 min.

8. Sample nomenclatures. “ WO_{3-x} ” and “WO-NWs” may cause misunderstanding and it may be better to be replaced by a more accurate term because tungsten oxides contain many nonstoichiometric compounds. According to the characterizations, nonstoichiometric $W_{18}O_{49}$ is confirmed.

Response: The XRD patterns of our prepared tungsten oxides nanowires are consisted with those of nonstoichiometric $W_{18}O_{49}$. However, from our XPS analysis, the actual amount of W^{5+} are much lower than the theoretical ratio of W^{5+}/W^{6+} in $W_{18}O_{49}$. From the high-resolution W 4f

XPS spectra of $W_{18}O_{49}$ -NWs and $W_{18}O_{49}$ -NWs/rGO-1% (Fig. S5). The molar ratio of W^{5+} in tungsten atoms is calculated to 11.5% and 17.1% for $W_{18}O_{49}$ -NWs and $W_{18}O_{49}$ -NWs/rGO-1%, respectively, which is lower than the 56% of $W_{18}O_{49}$. More interesting is that W^{5+} in $W_{18}O_{49}$ -NWs can be tuned by light irradiation as shown *in situ* W 4f XPS measurement (Fig. S6), and 24.6% W^{5+} is reached after 20 min full-spectrum light irradiation. However, considering the reviewer's suggestion, we revised the " WO_{3-x} " and WO-NWs to " $W_{18}O_{49}$ " and " $W_{18}O_{49}$ -NWs" in the whole revised manuscript.

Reviewers' Comments:

Reviewer #1:

Remarks to the Author:

In this revised manuscript, the authors clarified the novelty of the work, and answer my other questions on tech issues. I am happy to suggest acceptance for now.

Reviewer #2:

None

Reviewer #3:

Remarks to the Author:

The authors have addressed most of my concerns in the revised manuscript. The authors calculated the reaction activation barriers to investigate the contribution of plasmonic hot electron and photothermal effect on catalytic activity. They also newly performed several in-situ XPS and FTIR spectroscopic studies towards the reaction mechanism. Overall, I think that the revised manuscript becomes suitable for publication. Before the acceptance of this manuscript for publication, one concern about the photothermal effect should still be carefully considered. The authors stated that "the electrons transferred to rGO will have nonradiative decay to generate photothermal effect". Firstly, the plasmonic semiconductor itself (W18O49 in this study) can also contribute the robust photothermal effect. In the manuscript, the authors did not discuss this matter. Secondly, if we presume the same total amount of photogenerated electrons under the same reaction conditions, the electrons in defect band would become less after the partial electrons transfer to rGO. As the authors proposed that plasmonic hot electrons promote the C-O bond cleavage, it seems that this electron transfer to rGO is adverse to the whole reaction process.

Minor suggestion:

1. As an important result about in-situ FTIR study, I suggest that the authors add the experimental details into the main text rather than Supplementary Information.
2. The authors declared that SPR contributes to the promotion of C-O bond cleavage for activity enhancement. However, in Figure 5 about reaction mechanism, the authors draw that C-O bond is directly broken in transition state after step 2 and SPR has an effect on the following C-H cleavage and C=C formation. To avoid the misunderstanding, the authors should modify this illustration.

Response to the Referee

October 27, 2022

Reviewer: 3

Comments:

The authors calculated the reaction activation barriers to investigate the contribution of plasmonic hot electron and photothermal effect on catalytic activity. They also newly performed several in-situ XPS and FTIR spectroscopic studies towards the reaction mechanism. Overall, I think that the revised manuscript becomes suitable for publication. Before the acceptance of this manuscript for publication, one concern about the photothermal effect should still be carefully considered. The authors stated that “the electrons transferred to rGO will have nonradiative decay to generate photothermal effect”. Firstly, the plasmonic semiconductor itself ($W_{18}O_{49}$ in this study) can also contribute the robust photothermal effect. In the manuscript, the authors did not discuss this matter. Secondly, if we presume the same total amount of photogenerated electrons under the same reaction conditions, the electrons in defect band would become less after the partial electrons transfer to rGO. As the authors proposed that plasmonic hot electrons promote the C-O bond cleavage, it seems that this electron transfer to rGO is adverse to the whole reaction process.

Response: Thank you for reviewer reviewer’s comments. The fast nonradiative decay of hot electrons on plasmonic $W_{18}O_{49}$ nanowires plays the dominant contribution to the photothermal effect. Plasmonic heterostructures $W_{18}O_{49}$ -nanowires/rGO have more oxygen vacancies on $W_{18}O_{49}$ for stronger surface plasmon resonance, in this case, more hot electron generation and high photothermal effect are achieved. In addition, a portion of hot electrons and UV-excited electron at high energy level can transfer from $W_{18}O_{49}$ to rGO to have a nonradiative decay for thermal effect. Therefore, the higher surface temperature is observed on heterostructure $W_{18}O_{49}$ -nanowires/rGO (143.8 °C) than that of $W_{18}O_{49}$ -nanowires (94.2 °C).

For the second question, under full-spectrum light irradiation, the continuous UV-excited electrons on $W_{18}O_{49}$ -nanowires can be trapped on defect band. Therefore, in this case, even a portion of electron are transferred to rGO, the total electron density of $W_{18}O_{49}$ -nanowires is still kept at high level for catalysis. So, the electron transfer to rGO does not have adverse to reaction but can contribute part thermal effect for reaction.

Modification: We added the sentence “Those results indicate that $W_{18}O_{49}$ -NWs are the dominant contributors and that rGO also partially contributes to the high photothermal effect of plasmonic $W_{18}O_{49}$ -NWs/rGO-1%.” at p7 of the revised manuscript.

1. As an important result about in-situ FTIR study, I suggest that the authors add the experimental details into the main text rather than Supplementary Information.

Response: We have added the experimental details for in situ FTIR measurement in Methods section of the revised manuscript.

Modifications: The texts added in Methods section “In situ FTIR measurement. The in situ”

FTIR transmission spectra were measured by an in situ diffuse reflectance Fourier transform infrared spectrometer (Bruker Tensor II FTIR NEXUS). Before the experiment, the sample was degassed for 4 h at 150 °C. Then each sample was purged with nitrogen for 1 h to blow out all the gases adsorbed on the samples. After that, a mixed gas of 0.1 mL isopropanol and water vapor was flowed into the specimen chamber for another 30 min to ensure sorption equilibrium before irradiation. The FTIR transmission spectra of the sample were recorded every 2 min. Please see p12 of the revised manuscript.

2. The authors declared that SPR contributes to the promotion of C-O bond cleavage for activity enhancement. However, in Figure 5 about reaction mechanism, the authors draw that C-O bond is directly broken in transition state after step 2 and SPR has an effect on the following C-H cleavage and C=C formation. To avoid the misunderstanding, the authors should modify this illustration.

Response: Thank you for reviewer's comments. Above suggestion is useful to improve our manuscript. During the dehydration transition state of isopropanol, the C-O bond is broken firstly following with a fast proton remove from the remove of proton from C_β-carbon, leading to C=C formation for propylene on plasmonic W₁₈O₄₉-nanowires. So, we added the C-O cleavage process in the Figure 5 at p9 of the revised manuscript.

Modifications: The C-O bond cleavage process was added at step 2 in the revised Fig. 5.

The following two parts of texts have been modified:

1. Original: "Then, the dehydration transition state occurs, in which the C-O bond is activated for broken, and the incipient water molecule mediated β-elimination by accepting a proton from the C_β-carbon, donating a different proton back to the active site of the W₁₈O₄₉ catalyst."

Modified to "Then, the dehydration transition state occurs, in which the C-O bond is activated for cleavage at first state, and subsequently the incipient water molecule mediates β-elimination by accepting a proton from the C_β-carbon, which donates a different proton back to the active site of the W₁₈O₄₉ catalyst.^{23, 46}" Please see p10 of the revised manuscript.

2. Original: "Step 1: Isopropanol molecule diffuses to W₁₈O₄₉ surface for absorbed isopropanol (Isopropanol*), Step 2: Isopropanol* is activated for dehydration transition state, Step 3: C-O bond is broken for absorbed H₂O (H₂O*) and propylene (Propylene*) and Step 4: H₂O* and Propylene* are released from W₁₈O₄₉ surface."

Modified to Step 1: The isopropanol molecule diffuses to the W₁₈O₄₉ surface to be absorbed isopropanol (Isopropanol*); Step 2: Isopropanol* is activated for the dehydration transition state with C-O bond cleavage and water molecule elimination; Step 3: Isopropanol is dehydrated to be absorbed H₂O (H₂O*) and propylene (Propylene*); Step 4: H₂O* and Propylene* are released from the W₁₈O₄₉ surface. The blue cycle inset at right side shows full-spectrum-modulated plasmonic electron mechanism on W₁₈O₄₉." in caption of Fig. 5.

Having done our best to satisfactorily answer your comments, we hope that this revised version could meet the high standards of Nature Communications.

Yours sincerely,

Z. Lou and T. Guo
 Professor, Jinan University
 Associate Editor, IEEE Journal of Lightwave Technology
 Associate Editor, SCIENCE CHINA Information Sciences
 Technical Committee Chair, IEEE Instrumentation and Measurement Society